# System identification of neural systems: If we got it right, would we know?

## Abstract

Various artificial neural networks developed by engineers are now proposed as models of parts of the brain, such as the ventral stream in the primate visual cortex. The network activations are compared to recordings of biological neurons, and good performance in reproducing neural responses is considered to support the model's validity. This system identification approach, however, is only part of the traditional ways to develop and test models in the natural sciences. A key question is how much the ability to predict neural responses tells us. In particular, do these functional tests about neuron activation allow us to distinguish between different model architectures? We benchmark existing techniques to correctly identify a model by replacing brain recordings with known ground truth models. We evaluate the most commonly used identification approaches, such as a linear encoding model and centered kernel alignment. Even in the setting where the correct model is among the candidates, system identification performance is quite variable; it also depends significantly on factors independent of the ground truth architecture, such as stimuli images. In addition, we show the limitations of using functional similarity scores in identifying higher-level architectural motifs.

## 1 Introduction

Over the last two decades, the dominant approach for machine learning engineers in search of better performance has been to use standard benchmarks to rank networks from most relevant to least relevant. This practice has driven much of the progress in the machine learning community. A standard comparison benchmark enables the broad validation of successful ideas. Recently such benchmarks have found their way into neuroscience with the advent of experimental frameworks like Brain-Score (Schrimpf et al., 2020) and Algonauts (Cichy et al., 2021), where artificial models compete to predict recordings from real neurons in animal brains. Can engineering approaches like this be helpful in the natural sciences?

The answer is clearly yes: the "engineering approach" described above ranks models that predict neural responses better as better models of animal brains. While such rankings may be a good measure of absolute performance in approximating the neural responses, which on its own is valuable for various applications (Bashivan et al., 2019), it is an open question whether they are sufficient. In neuroscience, understanding natural intelligence at the level of the underlying neural circuits requires developing model systems that reproduce the abilities of their biological analogs while respecting the constraints provided by biology, including anatomy and biophysics (Marr & Poggio, 1976; Schaeffer et al., 2022). A model that reproduces neural responses well but turns out to require connectivity or biophysical mechanisms that are different from the biological ones is thereby falsified.

Consider the conjecture that the similarity of responses between model units and brain neurons allows us to conclude that brain activity fits better, for instance, a convolutional motif rather than a dense architecture. If this were true, it would mean that functional similarity over large data sets effectively constrains architecture. Then the need for a separate test of the model at the level of anatomy would become, at least in part, less critical for model validation. Therefore, we ask the question: could functional similarity be a reliable predictor of architectural similarity?

We describe an attempt to benchmark the most popular similarity techniques by replacing the brain recordings with data generated by various known networks with drastically different architectural motifs, such as convolution vs. attention. Such a setting provides a valuable upper bound to the identifiability of anatomical differences.

## 1.1 System identification from leaderboards

When artificial models are compared against common biological benchmarks for predictivity (Yamins & DiCarlo, 2016), models with the top score are deemed better models for neuroscience. As improvements to scores are made over time, ideally, more relevant candidates emerge. Nevertheless, if two artificial models with distinctly different architectures, trained on the same data, happen to be similar in reproducing neural activities (target model), then it would be impossible to conclude what accounts for the similarity. It can be biologically relevant *motifs from each architecture*, the properties of the *stimulus input*, or *similarity metric*. Such ambiguity is due to the many-to-one mapping of a model onto a leaderboard score. Our work shows that multiple factors play a role in representational similarities.

An interesting example is offered by Chang et al. (2021), which compares many different models with respect to their ability to reproduce neural responses in IT to face images and concludes that the 2D morphable model is best. Operations required in the specific model, such as correspondece and vectorization do not have an apparent biological implementation in terms of neurons and synapses. Nonetheless, it is perhaps not too surprising that the model can predict IT responses well, as there are multiple confounds besides the biological constraints, which affect the neural predictivity.

## 2 Related Work

While the analogy between neural network models and the brain has been well validated (Bashivan et al., 2019), the extent of this correspondence across multiple levels (Marr & Poggio, 1976) has been taken for granted. This assumed correspondence could be attributed to methodological limitations of evaluating such models simultaneously across all levels. Jonas & Kording (2017) investigated the robustness of standard analysis techniques in neuroscience with a microprocessor as a ground-truth model to determine the boundaries of what conclusions could be drawn about a known system. The presumption of correspondence could also be attributed to underappreciated variability from model hyperparameters (Schaeffer et al., 2022). In a similar spirit to Jonas & Kording (2017); Lazebnik (2002), we evaluate system identification on a known ground-truth model to establish the boundaries of what architectural motifs can be reliably uncovered. We perform our analysis under favorable experimental conditions to establish an upper bound.

As modern neural network models have grown more prominent in unison with the corresponding resources to train these models, pre-trained reference models have become more widely available in research (Wightman, 2019). Consequently, the need to compare these references along different metrics has followed suit. Kornblith et al. (2019); Morcos et al. (2018) explored using different similarity measures between the layers of artificial neural network models. Kornblith et al. (2019) propose various properties a similarity measure should be invariant such as orthogonal transformations and isotropic scaling while not invariant to invertible linear transformations. Kornblith et al. (2019) found centered kernel alignment (CKA), a method very similar to Representation Similarity Analysis (Kriegeskorte et al., 2008), to best satisfy these requirements. Ding et al. (2021) explored the sensitivity of methods like canonical correlation analysis, CKA, and orthogonal procrustes distance to changes in factors that do not impact the functional behavior of neural network models.

## 3 Background and Methods

The two predominant approaches to evaluating computational models of the brain are using metrics based on linear encoding analysis for neural predictivity and population-level representation similarity. The first measures how well a model can predict the activations of individual units, whereas the second metric measures how correlated the variance of internal representations is. We study the following neural predictivity scores consistent with the typical approaches: Linear Regression and Centered Kernel Alignment (CKA).

In computational neuroscience, we usually have a neural system (brain) that we are interested in modeling. We call this network a *target* and the proposed candidate model a *source*. Formally, for a layer with $p_1$ units in a source model, let $X \in \mathbb{R}^{n \times p_1}$ be the matrix of representations with $p_1$ features over $n$ stimulus images. Similarly, let $Y \in \mathbb{R}^{n \times p_2}$ be a matrix of representations with $p_2$ features of the target model (or layer) on the same $n$ stimulus images. Unless otherwise noted, we subsample 3000 target units to test an analogous condition as in biology, where recordings are far from exhaustive. Our analyses are partially dependent upon the target coverage, and we later examine the effect of increasing the number of target units.

### 3.1 ENCODING MODEL: LINEAR REGRESSION

Following the procedure developed by previous works (Schrimpf et al., 2020; Yamins et al., 2014; Conwell et al., 2021; Kar et al., 2019; Mitchell et al., 2008), we linearly project the feature space of a single layer in a source model to map onto a single unit in a target model (a column of $Y$). The linear regression score is the Pearson's correlation $r(\cdot, \cdot)$ coefficient between the predicted responses of a source model and the ground-truth target responses to a set of stimulus images.

$$\hat{\beta} = \text{argmin}_{\beta} ||Y - XS\beta||_F^2 + \lambda ||\beta||_F^2 \tag{1}$$

$$LR(X, Y) = r(XS\hat{\beta}, Y) \tag{2}$$

We first extract activations on the same set of stimulus images for source and target models. To reduce computational costs without sacrificing predictivity, we apply sparse random projection $S \in \mathbb{R}^{p_1 \times q_1}$ for $q_1 << p_1$, on the activations of the source model (Conwell et al., 2021). This projection reduces the dimensionality of the features to $q_1$ while still preserving relative distances between points (Li et al., 2006). Unlike principal component analysis, sparse random projection is a dataset-independent dimensionality reduction method. This removes any data-dependent confounds from our processing pipeline for linear regression and isolates dataset dependence into our variables of interest: linear regression and candidate model.

We apply ridge regression on every layer of a source model to predict a target unit using these features. We use nested cross-validations in which the regularization parameter $\lambda$ is chosen in the inner loop and a linear model is fitted in the outer loop. The list of tested $\lambda$ is $[0.01, 0.1, 1.0, 10.0, 100]$. We use 5-fold cross-validation for both inner and outer loops. As there are multiple target units, the median of Pearson's correlation coefficients between predicted and true responses is the aggregate score for layer-wise comparison between source and target models. Note that a layer of a target model is usually assumed to correspond to a visual area, e.g., V1 or IT, in the visual cortex. For a layer-mapped model, we report maximum linear regression scores across source layers for target layers.

### 3.2 CENTERED KERNEL ALIGNMENT

Another widely used type of metric builds upon the idea of measuring the representational similarity between the activations of two neural networks for each pair of images. While variants of this metric abound, including RSA or re-weighted RSA (Kriegeskorte et al., 2008; Khaligh-Razavi et al., 2017), we use CKA (Cortes et al., 2012) as (Kornblith et al., 2019) showed strong correspondence between layers of models trained with different initializations, which we will further discuss as a validity test we perform. We consider linear CKA in this work:

$$\text{CKA}(X, Y) = \frac{||Y^T X||_F^2}{||X^T X||_F ||Y^T Y||_F} \tag{3}$$

(Kornblith et al., 2019) showed that the variance explained by (unregularized) linear regression accounts for the singular values of the source representation. In contrast, linear CKA depends on the singular values of both target and source representations. Recent work (Diedrichsen et al., 2020) notes that linear CKA is equivalent to a whitened representational dissimilarity matrix (RDM) in RSA under certain conditions. We also call CKA a neural predictivity score because a target network is observable, whereas a source network gives predicted responses.

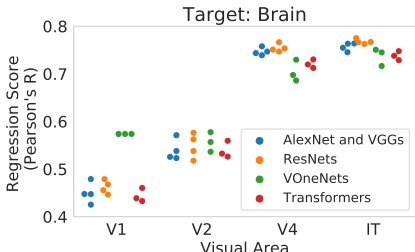

Figure 1: Linear regression scores of deep neural networks for brain activations in the macaque visual cortex. For V1, the top performing three models are in the VOneNet family (Dapello et al., 2020), which are explicitly designed to mimic the known properties of V1.

## 3.3 IDENTIFIABILITY INDEX

To quantify how selective predictivity scores are when a source matches the target architecture compared to when the architecture differs between source and target networks, we define an identifiability index as:

$$\text{Identifiability Index} = \frac{\text{Score(source = target)} - \text{Mean Score(source} \neq \text{target)}}{\text{Score(source = target)} + \text{Mean Score(source} \neq \text{target)}} \quad (4)$$

In brief, it is a normalized difference between the score for the true positive and the mean score for the true negatives. Previous works (Dobs et al., 2022; Freiwald & Tsao, 2010) defined selectivity indices in the same way in similar contexts, such as the selectivity of a neuron to specific tasks.

## 3.4 SIMULATED ENVIRONMENT

If a target network is a brain, it is essentially a black box, making it challenging to understand the properties or limitations of the comparison metrics. Therefore, we instead use artificial neural networks of our choice as targets for our experiments.

We investigate the reliability of a metric to compare models, mainly to discriminate the underlying computations specified by the model's architecture. We intentionally create favorable conditions for identifiability in a simulated environment where the ground truth model is a candidate among the source models. Taking these ideal conditions further, our target and source models are deterministic and do not include adverse conditions typically encountered in biological recordings, such as noise and temporal processing. We consider the following architectures:

**Convolutional Networks:** AlexNet (Krizhevsky et al., 2012), VGG11 (Simonyan & Zisserman, 2014), ResNet18 (He et al., 2016)

**Recurrent Networks:** CORnet-S (Kubilius et al., 2019)

**Transformer Networks:** ViT-B/32 (Dosovitskiy et al., 2020)

**Mixer Networks:** MLP-Mixer-B/16 (Tolstikhin et al., 2021)

These architectures are emblematic of the vision-based models used today. Each architecture has a distinct motif, making it unique from other models. For example, transformer networks use the soft-attention operation as a core motif, whereas convolutional networks use convolution. Recurrent networks implement feedback connections which may be critical in the visual cortex for object recognition (Kubilius et al., 2019; Kar et al., 2019). Moreover, mixer networks (Tolstikhin et al., 2021; Touvron et al., 2021; Melas-Kyriazi, 2021) uniquely perform fully-connected operations over image patches, alternating between the feature and patch dimensions.

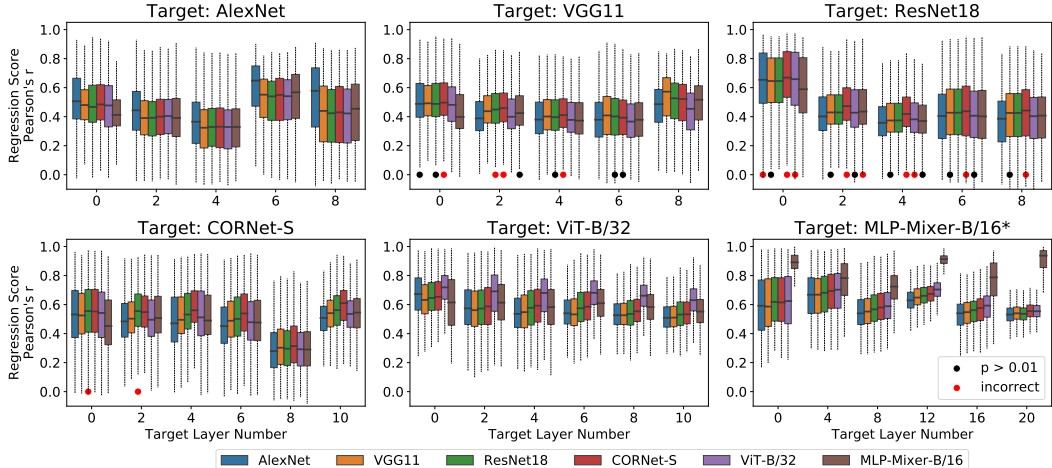

Figure 2: Linear regression scores for artificial neural networks shown in boxplots. We use different initialization seeds for source networks of the same architecture type as the target, except for MLP-Mixer-B/16 (marked with an asterisk), for which we test identical weights. Source networks marked with black dots indicate the correct architecture, identical to the target, does not outperform those networks with statistically significant difference ($p > 0.01$). Source networks marked with red dots indicate that median scores corresponding to those networks are higher than the median for the correct architecture. We compare the median as the ranking of source models is typically decided based on the aggregate median scores (Schrimpf et al., 2020).

## 4 RESULTS

### 4.1 DIFFERENT MODELS TRAINED ON A LARGE-SCALE DATASET REACH EQUIVALENT NEURAL PREDICTIVITY

We compare various artificial neural networks with publicly shared neural recordings in primates (Majaj et al., 2015; Freeman et al., 2013) via the Brain-Score framework (Schrimpf et al., 2020). Our experiments show that the differences between markedly different neural network architectures are minimal after training (Figure 1), consistent with the previous work (Schrimpf et al., 2020; Kubilius et al., 2019; Conwell et al., 2021). Previous works focused on the relative ranking of models and investigated which model yields the highest score. However, if we take a closer look at the result, the performance difference is minimal, with the range of scores having a standard deviation $< 0.03$ (for V2=0.021, V4=0.023, IT=0.016) except for V1. For V1, VOneNets (Dapello et al., 2020), which explicitly build in properties observed from experimental works in neuroscience, significantly outperform other models. Notably, the models we consider have quite different architectures based on combinations of various components, such as convolutional layers, attention layers, and skip connections. This suggests that architectures with different computational operations reach almost equivalent performance after training on the same large-scale dataset, i.e., ImageNet.

### 4.2 IDENTIFICATION OF ARCHITECTURES IN AN IDEAL SETTING

One potential interpretation of the result would be that different neural network architectures are equally good (or bad) models of the visual cortex. An alternative explanation would be that the method we use to compare models with the brain has limitations in identifying precise computational operations. To test the hypothesis, we focus on where underlying target neural networks are known instead of being a black box, as with biological brains. Specifically, we replace the target neural recordings with artificial neural network activations. By examining whether the candidate source model with the highest predictivity is identical to the target model, we can evaluate to what extent we can identify architectures with the current approach.

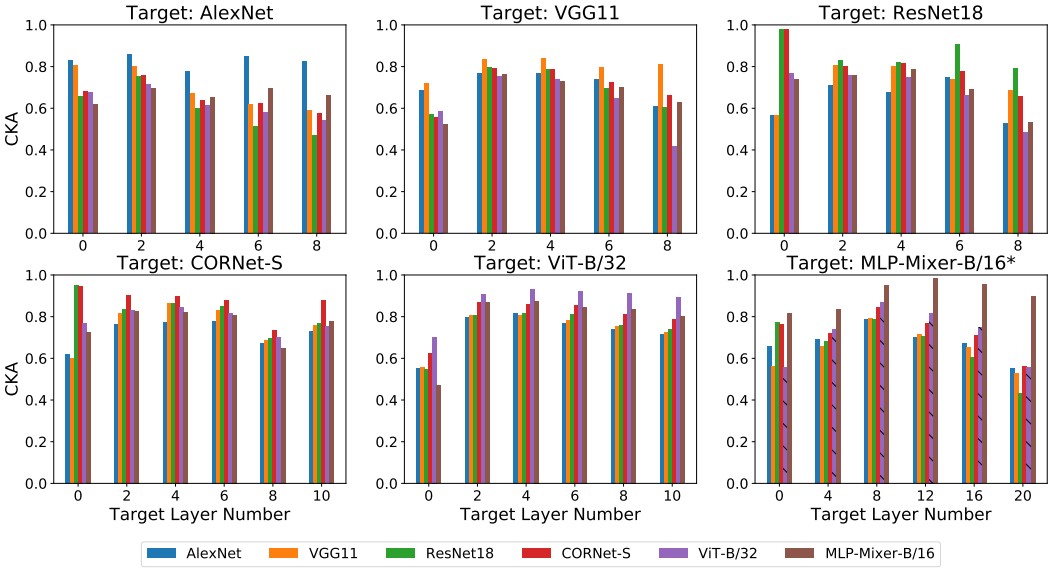

Figure 3: CKA for different source and target networks. The experimental setup is identical to Figure 2 besides using CKA instead of linear regression as the metric. As in Figure 2, when we test MLP-Mixer-B/16 as a target and the source network type matches the target, weights are identical. We show the results for MLP-Mixer-B/16 in bar plots with a pattern to indicate the difference from other targets.

### 4.2.1 LINEAR REGRESSION

We first compare various source models (AlexNet, VGG11, ResNet18, CORnet-S, ViT-B/32, MLP-Mixer-B/16) with a target network, the same architecture as one of the source models and is trained on the same dataset but initialized with a different seed. We test a dataset composed of 3200 images of synthetic objects studied in (Majaj et al., 2015) to be consistent with the evaluation pipeline of Brain-Score. The ground-truth source model will yield a higher score than other models if the model comparison pipeline is reliable. For most target layers, source networks with the highest median score are the correct network (Figure 2). However, strikingly, for several layers in VGG11, ResNet18, and CORnet-S the best-matched layers belong to a source model that is not the correct architecture. In other words, given the activations of VGG11, for instance, and based on linear regression scores, we would make an incorrect prediction that the system's underlying architecture is closest to a ResNet18.

In addition, because of our ideal setting, where an identical network is one of the source models, we expect to see a significant difference between matching and non-matching models. However, for some target layers in VGG11 and ResNet18, linear regression scores for the non-identical architectures, when compared with those for the identical one, do not show a significant decrease in predictivity based on Welch's t-test with $p < 0.01$ applied as a threshold (Figure 2). This result suggests that the identification of the underlying architectures of unknown neural systems is far from perfect.

### 4.2.2 CKA

Next, we examine another widely used metric, CKA, for comparing representations. Again, we compare different source models to a target model, also an artificial neural network. For the target models we tested, the ground-truth source models achieved the highest score (Figure 3). Still, some unmatched source networks lead to scores close to the matched networks, even for the target MLP-Mixer-B/16, where the source network of the same architecture type also has identical weights.

When applying CKA to compare representations, we subsample a set (3000) of target units to mimic the limited coverage of single-unit recordings. Assuming we can increase the coverage for future

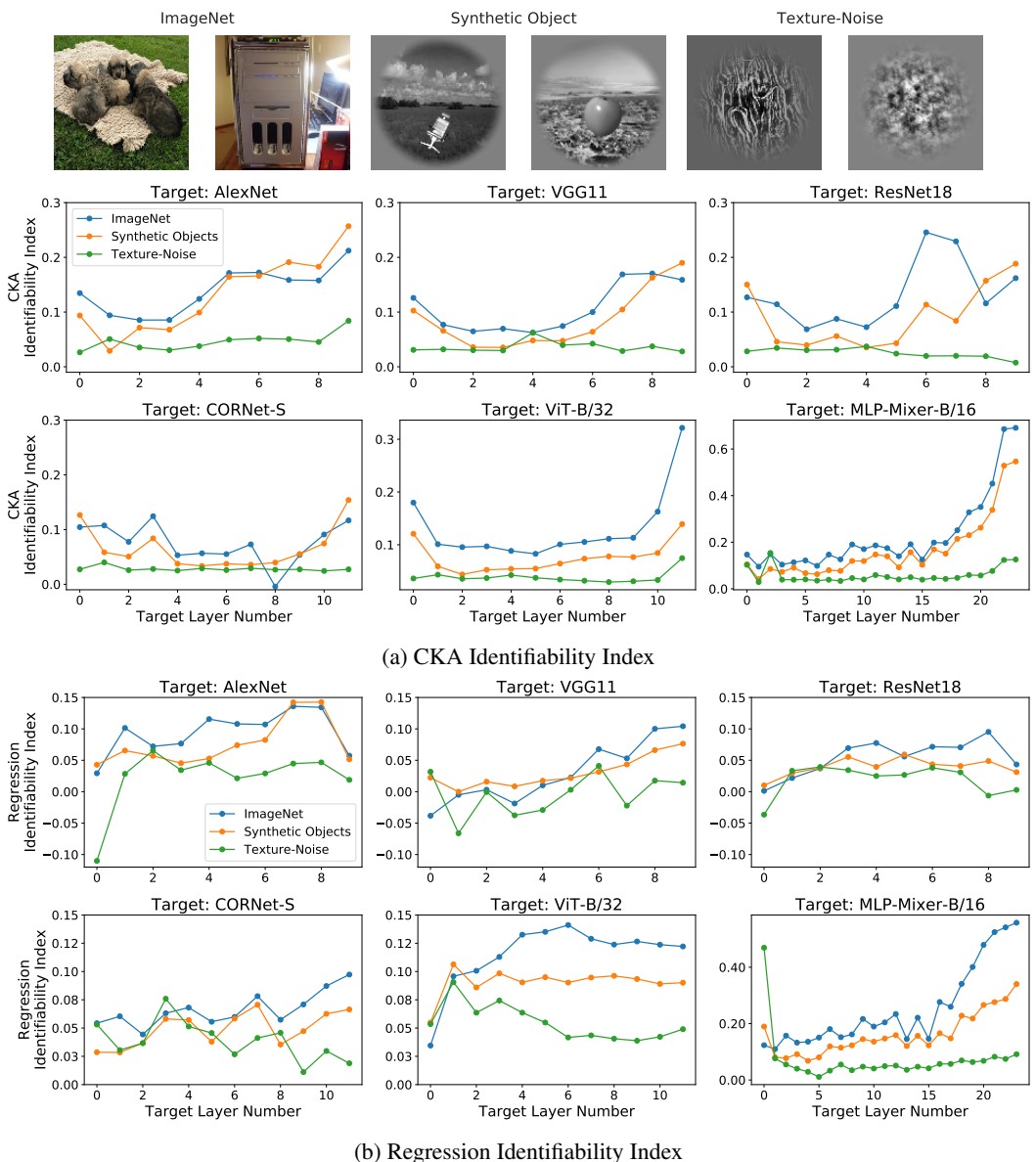

Figure 4: **Top** Sample images of each stimulus image type. **(a)** Identifiability index using CKA and **(b)** linear regression for different types of stimulus images and target networks.

experiments with more advanced measurement techniques, we test whether the identifiability improves if we include all target units. Additionally, methods similar to CKA, such as RSA, are often applied to interpret neural recordings, including fMRI, MEG, and EEG (Cichy & Oliva, 2020; Cichy & Pantazis, 2017), which can have full coverage of the entire brain or a specific region of interest. Therefore, we simulate such analyses by having all units in the targets. Overall, the ground-truth source models outperform the other source models with a significant margin (Figure 6. More experimental results are in the Appendix). This suggests system identification can be more reliable with more recording units and complete target coverage.

### 4.3 EFFECTS OF THE STIMULUS DISTRIBUTION ON IDENTIFIABILITY

A potentially significant variable overlooked in comparing computational models of the brain is the type of stimulus images. What types of stimulus images are suited for evaluating competing models?

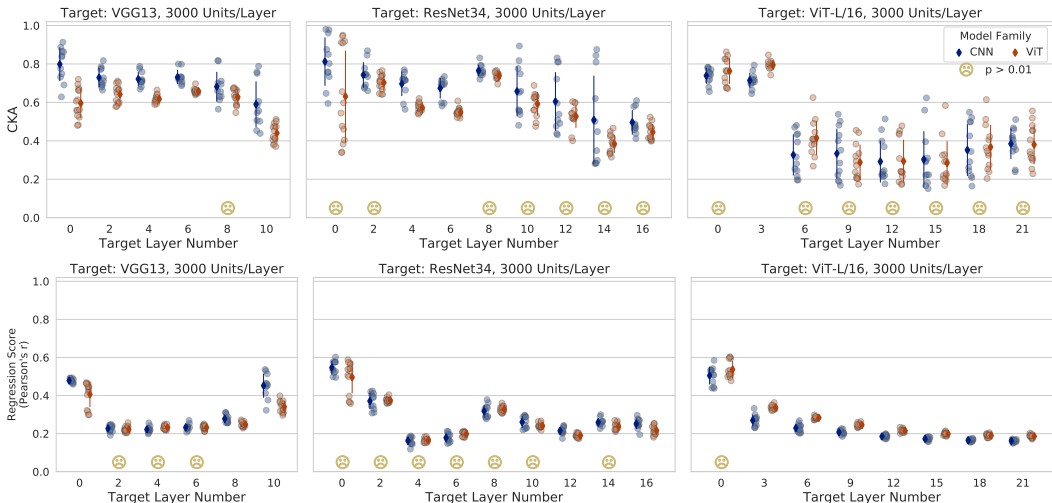

Figure 5: CNNs and ViTs of different architectural variants are compared with two CNNs and ViT target networks. Each data point is the maximum score of an architecture for corresponding target layers. Markers with darker shades indicate the mean score of the corresponding model class, and error bars are standard deviations. Frowning faces indicate that corresponding layers do not show a statistically significant difference between model classes.

In Brain-Score, stimulus images for comparing models of the high-level visual areas, V4 and IT, are images of synthetic objects (Majaj et al., 2015). In contrast, those for the lower visual areas, V1 and V2, are images of texture and noise (Freeman et al., 2013). To examine the effect of using different stimulus images, we test images of synthetic objects (3200 images), texture and noise (135 images), and ImageNet (3000 images), which are more natural images than the first two datasets.

In Figure 4, we analyze Identifiability Index for different stimulus images. More realistic stimulus images (i.e., synthetic objects and ImageNet) show higher identifiability than texture and noise images for all target models. For CKA, we observe identifiability increases with layer depth. Notably, even for early layers in target models, which would correspond to V1 and V2 in the visual cortex, texture and noise images fail to give higher identifiability. Also, between images of synthetic objects and ImageNet, ImageNet shows higher identifiability. As target models are trained on ImageNet, our results suggest that using stimuli images closer to the images that targets see will help identify architectures better.

It is important to note that the images of texture and noise we use in the experiment help characterize certain aspects of V1 and V2 in the previous work (Freeman et al., 2013). More specifically, the original work investigated the functional role of V2 in comparison with V1 by showing that naturalistic structure modulates V2. Although the image set plays an influential variable in a carefully designed experiment for a more targeted hypothesis, it does not translate as a sufficient test set for any hypothesis, such as evaluating different neural networks.

## 4.4 CHALLENGES OF IDENTIFYING KEY ARCHITECTURAL MOTIFS

Interesting hypotheses for a more biologically plausible design principle of brain-like models often involve key high-level architectural motifs. For instance, potential questions are whether recurrent connections are crucial in visual processing or, with the recent success of transformer models in deep learning, whether the brain similarly implements computations like attention layers in transformers. The details beyond the key motif, such as the number of layers or exact type of activation functions, may vary and be underdetermined within the scope of such research questions. Likewise, it is unlikely that candidate models proposed by scientists align with the brain at every level, from low-level specifics to high-level computation. Therefore, an ideal methodology for comparing models should help separate the key properties of interest while being invariant to other confounds.

Considering it is a timely question, with the increased interest in transformers as models of the brain in different domains (Schrimpf et al., 2021; Berrios & Deza, 2022; Whittington et al., 2021), we focus on the problem of identifying convolution vs. attention. We test 12 Convolutional Networks and 14 Vision Transformers of different architectures, and to maximize identifiability, we use ImageNet stimulus images. Note that an identical architecture with the target network is not included as a source network. Figure 5 shows that for most layers, mean CKA and regression scores are higher when there are correspondences between the target and source model classes than when there are not. However, three layers in ViT-L/16 (layers 9, 15, 21) show a higher mean for the CNN model family than for the ViT model family. Moreover, the architecture with the highest CKA score belongs to the CNN model family for three layers (layers 9-15).

Overall, we observe high inter-class variance for many target layers for both methods. For CKA, one layer in VGG13, 7 layers in ResNet34, and 7 layers in ViT-L/16, and for regression, three layers in VGG13, 7 layers in ResNet34, and one layer in ViT-L/16 do not show a statistically significant difference between the two model classes based on Welch's t-test with $p < 0.01$ used as a threshold. The significant variance among source models suggests that model class identification can be incorrect depending on the precise variation we choose, especially if we rely on a limited set of models.

## 5 DISCUSSION

Under idealized settings, we tested the identifiability of various artificial neural networks with differing architectures. We present two contrasting interpretations of model identifiability based on our results, one optimistic (Glass half full) and one pessimistic (Glass half empty).

**Glass half full:** Despite the many factors that can lead to variable scores, linear regression and CKA give reasonable identification capability under unrealistically ideal conditions. Across all the architectures tested, identifiability improves as a function of depth.

**Glass half empty:** However, system identification is highly variable and dependent on the properties of the target architecture and the stimulus data used to probe the candidate models. For architecture-wide motifs, like convolution vs. attention, scores overlap significantly across almost all layers. This indicates that such distinct motifs do not play a significant role in the score.

Our results suggest two future directions for improving system identification with current approaches: 1) Using stimuli images that are more natural, i.e., closer to the inputs to the target network (brain) in a natural setting. 2) With more neurons recorded in the brain, neural predictivity scores can be more reliable in finding the underlying architecture.

On the other hand, it is worthwhile to note that we may have reached close to the ceiling using neural predictivity scores for system identification. As an example, when our source network is AlexNet, its regression scores against the brain (Figure 1) are on par with, or slightly higher than, the scores against another AlexNet (Figure 2). In other words, based on the current methods, AlexNet predicts the brain as well as, if not better than, predicting itself. This observation is not limited to AlexNet but applies to other target networks. This fundamental limitation of present evaluation techniques, such as the linear encoding analysis used in isolation, emphasizes the need to develop new approaches beyond comparing functional similarities.

As we argued earlier, ranking models in terms of their agreement with neural recordings is the first step in verifying or falsifying a neuroscience model. Since several different models are very close in ranking, the next step – architectural validation – is the key. Furthermore, it may have to be done independently of functional validation and with little guidance from it, using standard experimental tools in neuroscience. A parallel direction is, however, to try to develop specially designed, critical stimuli to distinguish between different architectures instead of measuring the overall fit to data. As a simple example, it may be possible to discriminate between dense and local (e.g., CNN) network architectures by measuring the presence or absence of interactions between parts of a visual stimulus that are spatially separated.

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

# A APPENDIX

## A.1 MODEL DETAILS FOR SECTION 4.1: BRAIN-SCORE

Below is the full list of models tested on the benchmarks of Brain-Score as reported in Section 4.1. In addition to testing vision models pre-trained on ImageNet available from PyTorch's torchvision model package version 0.12, we test VOneNets that are pre-trained on ImageNet and made publicly available by the authors (Dapello et al., 2020). VOneNets are also a family of CNNs.

**Convolutional Networks:** AlexNet, VGG11, VGG13, VGG19, ResNet18, ResNet34, ResNet50, ResNet101, VOneAlexNet, VOneResNet50, VOnetCORnet-S

**Transformer Networks:** ViT-B/16, ViT-B/32, ViT-L/16, ViT-L/32

## A.2 MODEL DETAILS FOR SECTION 4.5: FINDING THE KEY ARCHITECTURAL MOTIF

For each target network reported in Section 4.5, namely VGG13, ResNet34, and ViT-L/16, below is the full list of source models tested to compare two model classes, CNN and transformer. For Tokens-to-token ViTs (T2T) (Yuan et al., 2021), we use models released by the authors. We use Twins Vision Transformers and a Visformer from timm library (Wightman, 2019). All other models are available from PyTorch's torchvision model package version 0.12. All models are pre-trained on ImageNet.

**Convolutional Networks**: AlexNet, VGG11, VGG13, VGG16, VGG13_bn, ResNet18, ResNet34, ResNet50, Wide-ResNet50_2, SqueezeNet1_0, Densenet121, MobileNet_v2

**Transfomer Networks:** ViT-B/16, ViT-B/32, ViT-L/16, ViT-L/32, T2T-ViT_t-14, T2T-ViT_t-19, T2T-ViT-7, T2T-ViT-10, Swin-B, Swin-S, Swin-T, Twins-PCPVT-Small, Twins-SVT-Small, Visformer-Small

## A.3 MODEL DETAILS: NUMBER OF LAYERS INCLUDED FOR EACH MODEL

Table 1

| Model | Number of Layers |
|---|---|
| AlexNet | 10 |
| CORnet-S | 12 |
| Densenet121 | 30 |
| MLP-Mixer-B16-224 | 24 |
| Mobilenet_v2 | 14 |
| ResNet18 | 10 |
| ResNet34 | 18 |
| ResNet50 | 18 |
| Squeezenet1_0 | 13 |
| Swin-B | 24 |
| Swin-S | 24 |
| Swin-T | 12 |
| T2T-ViT-10 | 13 |
| T2T-ViT-7 | 10 |
| T2T-ViT_t-14 | 17 |
| T2T-ViT_t-19 | 22 |
| Twins-PCPVT-Small | 16 |
| Twins-SVT-Small | 18 |
| VGG11 | 10 |
| VGG13 | 12 |
| VGG13-BN | 12 |
| VGG16 | 15 |
| Visformer-Small | 16 |
| ViT-B-16 | 12 |
| ViT-B-32 | 12 |
| ViT-L-16 | 24 |
| ViT-L-32 | 24 |
| Wide-ResNet502 | 18 |

## A.4 CKA RESULTS WHEN ALL TARGET UNITS ARE TESTED

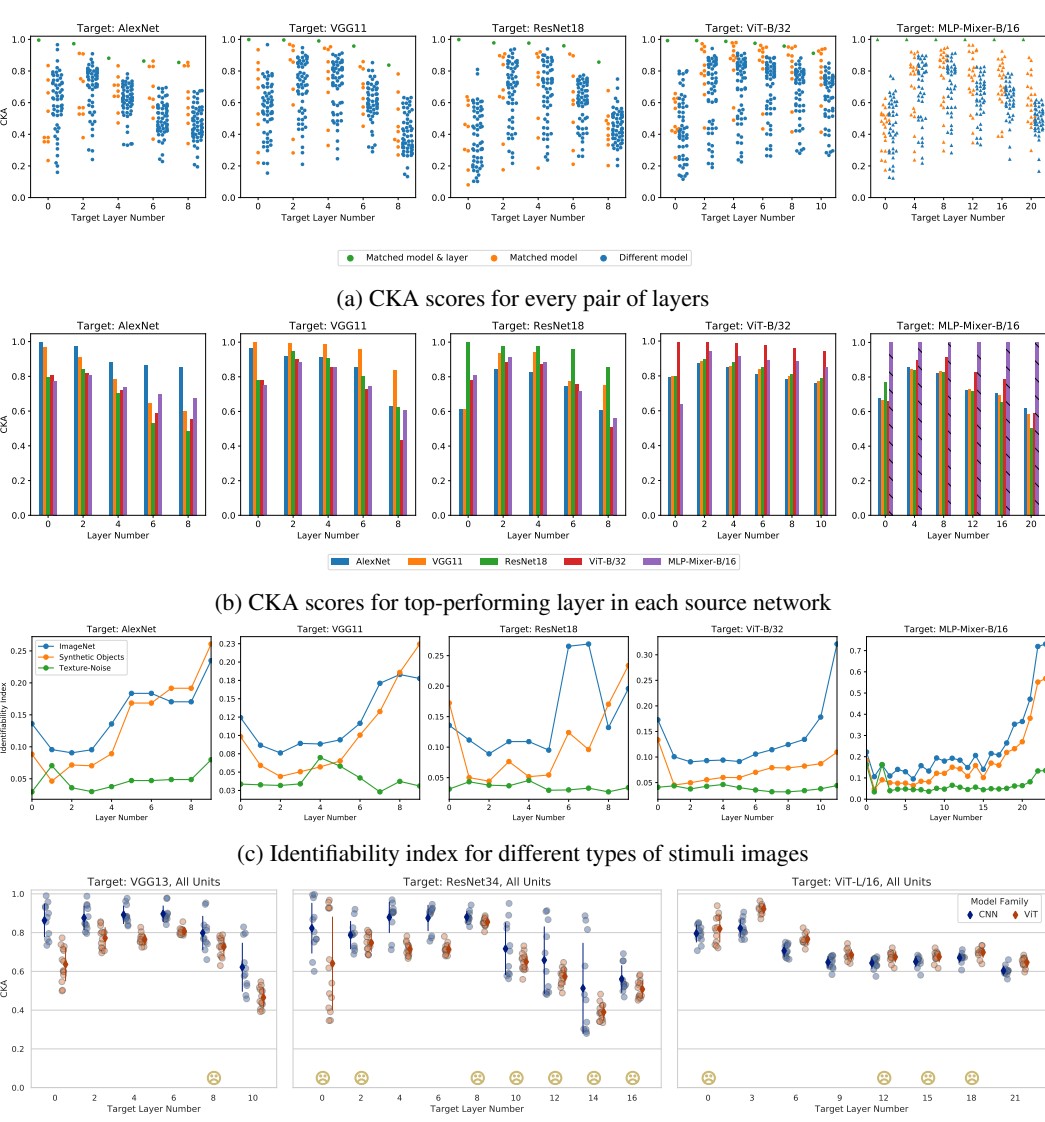

(a) CKA scores for every pair of layers

(b) CKA scores for top-performing layer in each source network

(c) Identifiability index for different types of stimuli images

(d) Comparing CKA scores for the model families CNN and ViT

Figure 6: We include all target units when computing CKA. Experimental setups are identical to Figures 3, 4, and 5 otherwise. See Section 4.2.2 for discussion. For the target MLP-Mixer-B/16 in (b), the source MLP-Mixer-B/16 has identical weights with the target; thus CKA is trivially 1.

