# OpenReview forum: "System identification of neural systems: If we got it right, would we know?"
_ICLR.cc/2023/Conference — Submitted to ICLR 2023_

### Official Review · Reviewer_eR5e · 2022-10-24

**Confidence:** 4
**Correctness:** 4
**Technical Novelty And Significance:** 3
**Empirical Novelty And Significance:** 3
**Recommendation:** 8

**Clarity, Quality, Novelty And Reproducibility:**

The paper is clear and easy to follow. The research question is
reasonably novel, and I couldn't find any inconsistency or
technical/correctness issue. The experimental procedures are
adequately described for reproducibility.

One minor issue I have found with the presentation quality is that the
labels and text on the plots is way too small. Please make sure that
the text is readable even when the document is visualized at its
"intended" dimension (as if printed on regular paper).

*Typos:*
- page 3, paragraph below equation 2: "principle components" (should
  be "principal").

**Strength And Weaknesses:**

## Strengths
1. The idea of this paper will be of interest to the iclr
   community. Regardless of the issues I have with some of the
   conceptual framing of this paper (see below, point 1 in weaknesses), I
   believe it is useful to get a rough sense of what one can expect even in a best case scenario, when using these similarity/alignment metrics.
2. The range of networks analyzed seems appropriate and the results
   are reported in a clear way.
3. The paper conclusions are given in a balanced way, presenting both
   a "glass half full" and a "glass half empty" perspective.

## Weaknesses
1. Several passages in the text seem to imply that comparing deep net
   and brain representations of certain stimuli must be the same as
   testing the hypothesis that information processing in the brain
   follows the same patterns/motifs as the deep net. While this is
   certainly true of some existing works in the literature, it is not
   true of all of them. Indeed, one could appreciate having a good
   model of neural activations even from a purely phenomenological
   perspective. Just to make one example, in the Bashivan paper cited
   among the references, the deep net allows the experimenters to
   design stimuli that can drive target neurons very strongly. This
   would still have value even if the model brought no insight
   whatsoever about information processing in visual cortex, as it
   would enable new kinds of experiments. Now, I understand that the
   present paper focuses on system identification (so exactly the
   "hypothesis testing" scenario), but it would be good to mention
   somewhere that system identification is not the only possible
   motivation behind modeling the ventral stream/brain circuits using
   deep networks.
2. The value of λ (the regularization parameter for the linear
   regression) is fixed at a hand-picked value, rather than selected
   in a systematic way, for instance by cross-validation. Other values
   of λ are explored in the supplement. This seems like a suboptimal
   way of performing a systems identification analysis. The results of
   the paper would be strengthened if λ was not picked by hand.
3. I found it interesting to learn how identification reliability
   increases with increasing population size. The main analyses were
   done with populations of 3000 neurons, an arbitrary number. It is
   possible that using a larger number (still compatible with the
   upper limits of what is possible experimentally) would have yielded
   somewhat different results. This is especially true for some of the
   experiments that are already borderline, like the results in figure
   2, where a lot of emphasis is given to the fact that 2 layers out
   of 5 in one network out of five tested do not have the ground-truth
   model as the best match. In this light, it would be good to warn
   the reader up front, when specifying the size of the populations
   used (page 3, top) that the results are dependent on this choice.

**Summary Of The Paper:**

This paper considers two approaches typically used in comparing deep
nets and brain recordings, linear regression and CKA, and explores
their behavior when used to measure the similarity of different neural
networks. This is used to highlight under which conditions (and in
what sense) these methods allow to identify common patterns between
information processing systems.

**Summary Of The Review:**

This paper provides some data about the behavior of common
similarity/alignment measures used to compare deep nets and neural
recordings. Although descriptive in nature, it will provide a useful
reference for a sizeable subcommunity at iclr.

---

> ### Author Response · Authors · 2022-11-19
> **Response to Reviewer eR5e**
>
> Thank you for the helpful review. You can find our updates to the manuscript highlighted in purple.
>
> > Now, I understand that the present paper focuses on system identification (so exactly the "hypothesis testing" scenario), but it would be good to mention somewhere that system identification is not the only possible motivation behind modeling the ventral stream/brain circuits using deep networks.
>
> Yes, we agree there can be other purposes for having models that predict neural responses. We mention this point a couple of times in the introduction and discussion but expanded it in the introduction with the relevant citation.
>
> > The value of λ (the regularization parameter for the linear regression) is fixed at a hand-picked value, rather than selected in a systematic way, for instance by cross-validation
>
> We re-ran experiments using nested cross-validation and updated Figure 2 and 4 accordingly. We also updated our explanation in the method Section 3.1. These experiments are computationally expensive jobs, so re-running Figure 5 regression experiments has not been completed. We will update the figure with those results once the run is complete. However, considering how scores are close in an ideal setting (Figure 2), where one of the source networks exactly matches the target, we do not expect that the new results for Figure 5 will show a significant gap between the two model classes.
>
> > The main analyses were done with populations of 3000 neurons, an arbitrary number. It is possible that using a larger number (still compatible with the upper limits of what is possible experimentally)... In this light, it would be good to warn the reader up front, when specifying the size of the populations used (page 3, top) that the results are dependent on this choice.
>
> Based on your feedback, we added explicit comments about the size of populations in Section 3. On a related note, our test condition of including 3000 neurons per layer can be viewed as already exceeding the relative population size (compared to the total number of neurons) recorded in the brain. "Neuron densities vary across and within cortical areas in primates" (Collins et al., 2010) reports in a primate visual cortex there are approximately 35 million neurons in V1, with decreasing number of neurons in the higher visual areas with 1.6 million neurons in MT. Single neuron recording sites are generally on the order of a few hundreds when different visual areas are combined. We approximate the number of neural recording sites as 400 neurons assuming 100 sites for each V1, V2, V4, and IT are all combined in one visual area to maximize the coverage. As the number of units roughly ranges from 800K (early layer of ResNet) to 4096 (later layer of AlexNet), if we compute the same ratio for an artificial neural network, the number of units would range from 1 - 200 for a single layer, which is smaller than 3000 neurons.
>
> > One minor issue I have found with the presentation quality is that the labels and text on the plots is way too small.
>
> We increased the figure size and the font size for the texts on the figures.
>
> > Typo
>
> Thanks for pointing this out. We fixed the typo.

---

> > ### Author Response · Authors · 2022-12-12
> > **Figure 5 revision**
> >
> > > The value of λ (the regularization parameter for the linear regression) is fixed at a hand-picked value, rather than selected in a systematic way, for instance by cross-validation
> >
> > We have updated Figure 5 ridge regression results to use nested cross-validations as you suggested (anonymized link: https://docs.google.com/document/d/e/2PACX-1vRs3E3xX0V8xJrBT6eMyZstLCXU19XXt1pJvG9BhHrgiAo0Gw_Mj2sFTnbyU9URuyT82CyaKRRmAfxI/pub). The overall results are consistent with previous ones, using a fixed ridge regularization parameter, and do not change our conclusion that the scores for the two model classes overlap significantly.

---

### Official Review · Reviewer_ngXd · 2022-10-24

**Confidence:** 5
**Correctness:** 3
**Technical Novelty And Significance:** 2
**Empirical Novelty And Significance:** 2
**Recommendation:** 3

**Clarity, Quality, Novelty And Reproducibility:**

The clarity, quality, and reproducibility are solid.
The approach is not inherently novel (it is close to Kornblith, Norouzi, Lee & Hinton (2019)), but the significance can be improved by performing additional analyses that better reveal the reasons for the limitations of the current methods.

**Strength And Weaknesses:**

Disclaimer: I reviewed a very similar version of this work at NeurIPS 2022. The authors have addressed some of my concerns in the current work but several of my major concerns remain and I will summarize these below, along with some new ones in this version.

Strengths:
- Investigates a problem that is becoming increasingly important as the popularity of the research area of relating representations in machines and brains grows
- The experiment that examines the effect of different types of images used for evaluating the alignment is novel as far as I know, and important for this subfield.

Weaknesses:
Major:
1. The ridge regularization parameter is fixed at 1. This parameter needs to be optimized with nested cross-validation in order to get a fairer estimate of the generalization performance. This is important since one of the points the authors are trying to make is that CKA is (slightly) better than ridge regression at system identification, and also that the difference between the performances for the correct and incorrect systems is small.
2. The takeaways from the work are not clear enough. The results suggest that system identification is possible in most cases with the current approaches in an ideal setting. How generalizable are these results to neural recordings? The authors can do more to investigate this question since it is the main motivation behind the work. In addition, the work tries to interpret the results pessimistically (though this is toned-down a lot from the previous version), but this further muddles the message. It is furthermore not clear whether any significance testing was done for the results (e.g. "However, for some target layers in AlexNet and ResNet18, although the layer with the highest score may be the matching layer in the same architecture, linear regression scores for other source models do not show a significant decrease in predictivity" is this statement based on a significance test?)
3. All of the figures are illegible due to size, small fonts, and blur.

Minor:
- "Following the procedure developed by previous works (Schrimpf et al 2020, Yasmins et al 2014, Conwell et al 2021).." The procedure for training encoding models for predicting brain recordings dates back much further than the cited works. See Kay et al. 2008 Nature (https://www.ncbi.nlm.nih.gov/pmc/articles/PMC3556484/), and Mitchell et al. 2008 Science (https://pubmed.ncbi.nlm.nih.gov/18511683/).

Suggestion:
- "One potential interpretation of the result would be that different neural network architectures are equally good (or bad) models of the visual cortex. An alternative explanation would be that the method we use to compare models with the brain has limitations in identifying precise computational operations. To test this hypothesis,.." -- Yet another alternative explanation (which I believe is actually more likely) is that even though different source models explain similar percent variance in the target model, this variance may be different from model to model (especially when comparing different layers of models). It is not uncommon nowadays to do a variance partitioning analysis (both for encoding models and for RSA/CKA) to better understand the contributions to predictive performance. One suggestion is to partition the explained variance in the target system by the different candidate source systems, which may provide more insight into why different source systems achieve good prediction performance for the same target system.

**Summary Of The Paper:**

This work takes a closer look at the prevalent tools in computational cognitive neuroscience today that are used to relate representations of deep learning models to representations in the brain. These tools are encoding models and a form of representational similarity analysis (CKA). The main question in this work goes along one of the several directions in this increasingly popular research area, which is to compare different deep learning architectures in terms of their alignment with brain recordings and conclude that the best performing one must be closer to the actual brain system. This specific work examines whether these comp neuro tools are indeed able to identify the correct target system, in the case that the target system is known. For this purpose, the work investigates the alignment (both in terms of regression scores and representational similarity) among various deep learning models. It reports that even though both encoding models and CKA are able to identify the correct system in most cases, the differences between the correct and incorrect source systems is small. Furthermore, the work also examines the effect of different types of images used for evaluating the alignment and finds that more naturalistic images lead to higher identifiability.

**Summary Of The Review:**

While this work investigates an important and timely problem, it does not yet go deep enough to warrant a top-tier ML publication in my opinion. There are several good directions that this work has started (e.g. investigate the effect of training data distribution of the source and possibly target domain) and I encourage the authors to further pursue them.

---

> ### Author Response · Authors · 2022-11-19
> **Thank you for the feedback**
>
> Thank you for reviewing our paper again. We greatly appreciate your suggestions and constructive feedback for both reviews. Therefore, it was to our surprise that the review score dropped compared to the previous submission. If this was not intended, we would appreciate it if you can re-evaluate our work. As you noted, we also incorporated feedback from the previous review.

---

> ### Author Response · Authors · 2022-11-19
> **Response to Reviewer ngXd**
>
> Thanks for your comments again. You can find our updates to the manuscript highlighted in purple.
>
> > The ridge regularization parameter is fixed at 1. This parameter needs to be optimized with nested cross-validation in order to get a fairer estimate of the generalization performance.
>
> We re-ran experiments using nested cross-validation and updated Figure 2 and 4 accordingly. We also updated our explanation in the method Section 3.1. These experiments are computationally expensive jobs, so re-running Figure 5 regression experiments has not been completed. We will update the figure with those results once the run is complete. However, considering how scores are close in an ideal setting (Figure 2), where one of the source networks exactly matches the target, we do not expect that the new results for Figure 5 will show a significant gap between the two model classes.
>
> > How generalizable are these results to neural recordings?
>
> Instead of directly adding noise, what we investigate in the study is the condition where we do not have full coverage of neural recordings, by subsampling target units. Without making too many assumptions, this condition mimics the limited neural recordings.
>
> We considered adding noise to target network activations to mimic the noisy neural firing, but it is not trivial to decide what assumptions about the type and degree of noise are reasonable. We can try adding some gaussian noise but regardless of the results, it would be questionable whether the outcome is relatable to biology. We would appreciate it if you have further suggestions in this direction.
>
> > It is furthermore not clear whether any significance testing was done for the results
>
> Based on the feedback, we updated Figure 2 with boxplots to show the full score distribution for target units. We also added results for performing statistical testings to compare scores with the correct ones. We highlighted the changes in the text (Section 4.2.1). Our updated results provide further evidence that scores for the correct cases often do not significantly outperform those for incorrect ones.
>
> > All of the figures are illegible due to size, small fonts, and blur.
>
> We made Figures 2, 3, and 4 larger, and increased the text size on all figures. Note that data points in Figure 5 are not blurry but drawn as partially transparent to display partly overlapping scores.
>
> > "Following the procedure developed by previous works (Schrimpf et al 2020, Yasmins et al 2014, Conwell et al 2021).." The procedure for training encoding models for predicting brain recordings dates back much further than the cited works. See Kay et al. 2008 Nature (https://www.ncbi.nlm.nih.gov/pmc/articles/PMC3556484/), and Mitchell et al. 2008 Science (https://pubmed.ncbi.nlm.nih.gov/18511683/).
>
> We agree that those works are relevant. We added the references.
>
> >  The approach is not inherently novel (it is close to Kornblith, Norouzi, Lee & Hinton (2019)), but the significance can be improved by performing additional analyses that better reveal the reasons for the limitations of the current methods.
>
> Although our approach is relevant to the previous work Kornblith, Norouzi, Lee & Hinton (2019) in the sense that we compare representations of artificial neural networks, and indeed we refer to the work, our research objectives and questions are novel, different from previous works. In the previous work, various architectures were not compared to make inferences about the target architecture, and analyses are mainly examining layerwise correspondence. Furthermore, we use measurement settings and stimuli relevant to real-world neuroscience experiments. The question we ask is explicitly motivated by the computational neuroscience field where the architecture of target networks is unknown.
>
> > One suggestion is to partition the explained variance in the target system by the different candidate source systems
>
> Thanks for the suggestion. Based on your comments, we are currently investigating in this direction to expand on what makes system identification hard based on functional tests.

---

> > ### Author Response · Authors · 2022-12-12
> > **Figure 5 revision**
> >
> > > The ridge regularization parameter is fixed at 1. This parameter needs to be optimized with nested cross-validation in order to get a fairer estimate of the generalization performance.
> >
> > We have updated Figure 5 ridge regression results to use nested cross-validations as you suggested (anonymized link: https://docs.google.com/document/d/e/2PACX-1vRs3E3xX0V8xJrBT6eMyZstLCXU19XXt1pJvG9BhHrgiAo0Gw_Mj2sFTnbyU9URuyT82CyaKRRmAfxI/pub).
> > The overall results are consistent with previous ones, using a fixed ridge regularization parameter, and do not change our conclusion that the scores for the two model classes overlap significantly.

---

### Official Review · Reviewer_RWEG · 2022-10-25

**Confidence:** 3
**Correctness:** 4
**Technical Novelty And Significance:** 1
**Empirical Novelty And Significance:** 2
**Recommendation:** 3

**Clarity, Quality, Novelty And Reproducibility:**

The paper is clearly written and the architecture parameters are fully reported. While the paper has some novel aspects, this type of work has been already done, and new contributions need more extensive examinations (e.g. see the references above).



**Strength And Weaknesses:**

Strength: This paper tackles an important problem in computational neuroscience and raises a very concerning issue regarding using deep networks for (neuro)science. It is clearly written and gives a nice overview of the literature on using neural networks for neuroscience.

Weakness:
My main concern is that the authors did not really provide a solution (or proof that there is none) for the raised issue. It is also not clear that the issues arise because of the measurements themselves or the authors' specific statistical choices such as reporting the median for correlations. I think a more in-depth analysis is needed to flesh out issues of using artificial architectures in neural recording analysis. For example, I think it is necessary to show how the measures and conclusions change with the increase in neural sample size, especially because increasing neural recording size is suggested as one of the solutions. In addition, as mentioned in the paper there exist several measures in these types of analysis. Given that the experiments are all simulations on very common networks, I think trying other measures such as SVCCA and SVD is necessary (see "SVCCA: Singular Vector Canonical Correlation Analysis for Deep Learning Dynamics and Interpretability" by Raghu et al, NeurIPS 2017 and "insights on representational similarity in neural networks with canonical correlation" by Morcos et al, NeurIPS 2018).
Finally, recurrent networks should be examined too especially because of feedback connections in the visual cortex.

**Summary Of The Paper:**

This paper challenges the interpretability of common neural recording analysis methods with deep artificial networks, by simulating such methods on artificial networks themselves. First, it shows that several architectures with totally different components are very similar in commonly used prediction measures when tested on a neural recording benchmark data set. Second, it shows that these common measures can not distinguish the ground truth even when it is an artificial neural network. Importantly, this simulation is done with a sample size of target neurons analogous to neuroscience experiments. Moreover, the authors show high variability in results when the stimulus type (family of the input images) changes. The authors briefly suggested two ways to overcome the interpretability issue, i.e. using more natural stimuli, and recording from more neurons.

**Summary Of The Review:**

The paper is clear and tackles an interesting problem but needs further experiments given the history of this type of work in the DL community.

---

> ### Author Response · Authors · 2022-11-19
> **Response to Reviewer RWEG**
>
> Thank you again for the helpful comments. You can find our updates to the manuscript highlighted in purple.
>
>
> > My main concern is that the authors did not really provide a solution (or proof that there is none) for the raised issue.
>
> Our main objective is to understand the capabilities and limitations of functional tests, which are becoming popular in neuroscience to validate models of the brain. That said, we also provide suggestions for improving functional tests by using more natural stimuli and recording more neural sites. However, we wanted to provide a balanced (or more realistic) view that even with these improvements, if the tested hypotheses are on high-level motifs, they are not easily verified with functional tests. Finally, we discuss we still need the traditional approaches of conducting hypothesis-driven experiments or using specially designed stimuli for those experiments to complement the functional tests.
>
> > It is also not clear that the issues arise because of the measurements themselves or the authors' specific statistical choices such as reporting the median for correlations.
>
> We updated Figure 2 with boxplots to show the distribution of correlation scores for target units. We also report results for performing statistical tests to compare scores with the correct ones. Our updated results provide further evidence that scores for the correct cases often do not outperform those for incorrect ones.
>
> > I think it is necessary to show how the measures and conclusions change with the increase in neural sample size, especially because
> increasing neural recording size is suggested as one of the solutions.
>
> We investigate the effect of increasing the neural sample size by including all target units when using CKA (Figure 6 in the Supplementary and referenced in section 4.2.2). For an ideal setting where one of the source networks matches the target exactly, we observe bigger differences between correct and incorrect cases (better identifiability). Comparing the key architectural motifs, namely convolution vs. attention, is improved but still limited.
>
> Due to the computational feasibility limit, we have not tested including more than 3000 target units for regression. On a related note, our test condition of including 3000 neurons per layer can be viewed as already exceeding the relative population size (compared to the total number of neurons) recorded in the brain. "Neuron densities vary across and within cortical areas in primates" (Collins et al., 2010) reports in a primate visual cortex there are approximately 35 million neurons in V1, with decreasing number of neurons in the higher visual areas with 1.6 million neurons in MT. Single neuron recording sites are generally on the order of a few hundreds when different visual areas are combined. We approximate the number of neural recording sites as 400 neurons assuming 100 sites for each V1, V2, V4, and IT are all combined in one visual area to maximize the coverage. As the number of units roughly ranges from 800K (early layer of ResNet) to 4096 (later layer of AlexNet), if we compute the same ratio for an artificial neural network, the number of units would range from 1 - 200 for a single layer, which is smaller than 3000 neurons.
>
> > Given that the experiments are all simulations on very common networks, I think trying other measures such as SVCCA and SVD is necessary (see "SVCCA: Singular Vector Canonical Correlation Analysis for Deep Learning Dynamics and Interpretability" by Raghu et al, NeurIPS 2017 and "insights on representational similarity in neural networks with canonical correlation" by Morcos et al, NeurIPS 2018).
>
> We agree that the suggested works are relevant and added the reference to Section 2. Although we are currently running experiments using SVCCA, we do not particularly expect to see better identification capability, given the inferior accuracy of SVCCA to CKA for identifying corresponding layers between the same network with different initialization seeds ("Similarity of Neural Network Representations Revisited" by Kornblith et al., 2019).
>
> > Finally, recurrent networks should be examined too especially because of feedback connections in the visual cortex.
>
> We agree that recurrent networks are interesting candidate models to include in our analysis. We ran new analyses on CORnet-S, which is a recurrent network inspired by the organization of the macaque visual cortex and shown to have a high neural predictivity score (Kubilius et al., 2019). Figures 2, 3, and 4 are updated accordingly.

---

### Decision · Program_Chairs · 2023-01-20

**Decision:**

Reject

**Justification For Why Not Higher Score:**

Some of the experiments required to accept this paper are underway - I do think this is an important question and parts of it are answered well, so I could see this being bumped up (though I think that's unlikely given the two 3s)

**Justification For Why Not Lower Score:**

n/a

**Metareview: Summary, Strengths And Weaknesses:**

This paper attempts to answer an interesting question: if we had a NN that matched a biological system, would we be able to tell?  Through a series of well controlled studies, the answer seems to be yes, some of the time, in close-to-ideal scenarios.

The reviewers pointed out strengths in the paper: the authors agreed that the paper was clear and that the issue handled by the paper was important and timely.

The reviewers also flagged some weaknesses: the results depend pretty strongly on the design choices the authors made.  Those design choices need to be fully explained, or verified to be ideal empirically. The authors have emabarked on this mission, but the results are not yet in.  The reviewers also wanted the authors to more clearly articulate the "so what" of this paper -- what should we do differently in light of these results.